# UNISPEAKER: A UNIFIED SPEECH GENERATION MODEL FOR MULTIMODALITY-DRIVEN VOICE CONTROL

## ABSTRACT

Recent advancements in zero-shot speech personalized generation have brought synthetic speech increasingly close to the realism of target speakers' recordings, yet multimodal voice creation remains on the rise. In various scenarios, individuals often seek to control and create voice characteristics through different voice description modalities. To address the limitations in both the versatility and performance of voice control found in previous methods, this paper introduces UniSpeaker, a unified multimodality-driven speech generation model that integrates face images, text descriptions, voice attribute descriptions, and reference speech for comprehensive voice control and creation. Specifically, we propose a unified voice aggregator based on KV-Former, applying soft contrastive loss to map diverse voice description modalities into a shared voice space, ensuring that the generated voice aligns more closely with the input descriptions. In addition, multimodal voice control is incorporated within a large-scale speech generation framework, employing self-distillation to enhance voice disentanglement. To evaluate multimodality-driven voice control, we build the first multimodality-based voice control (MVC) benchmark, focusing on voice suitability, voice diversity, and speech quality. UniSpeaker is evaluated across five tasks using the MVC benchmark, and the experimental results demonstrate that UniSpeaker outperforms previous modality-specific models. Speech samples are available at https://UniSpeaker.github.io.

**UniSpeaker**

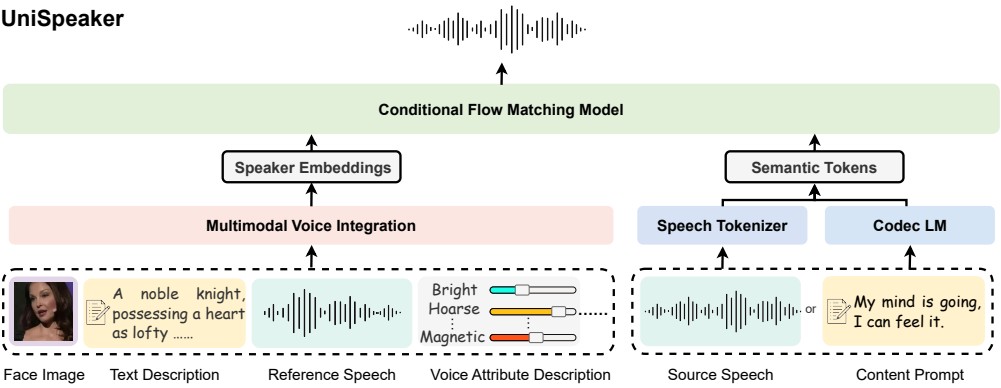

Figure 1: The overview of UniSpeaker, where speaker embeddings and semantic tokens serve as intermediate representations. Speaker embeddings are used to control the voice characteristics of generated speech and can be derived from various voice description modalities. Semantic tokens can be generated from source speech or text, corresponding to the speech-to-speech conversion and text-to-speech synthesis, respectively. Codec LM represents the text-to-token language model.

Table 1: Comparison between UniSpeaker and previous studies on multimodality-driven voice control tasks. TTS stands for text-to-speech synthesis. SSC stands for speech-to-speech conversion, preserving both the content and prosody of source speech.

| Model | Voice Description Modality | | | | Speech Generation Approaches | |
|---|---|---|---|---|---|---|
| | Speech | Text | Face | Attribute | TTS | SSC |
| PromptSpeaker(Zhang et al., 2023) | ✓ | ✓ | | | ✓ | |
| Promptttts++ (Shimizu et al., 2024) | ✓ | ✓ | | | ✓ | |
| PromptVC (Yao et al., 2024a) | ✓ | ✓ | | | | ✓ |
| HybridVC (Niu et al., 2024) | ✓ | ✓ | | | | ✓ |
| Imaginary Voice (Lee et al., 2023) | | | ✓ | | ✓ | |
| Synthe-sees (Park et al., 2024) | ✓ | | ✓ | | ✓ | |
| 3D-face (Yang et al., 2023) | ✓ | | ✓ | | ✓ | |
| FaceVC (Lu et al., 2021) | ✓ | | ✓ | | | ✓ |
| SP-FaceVC (Weng et al., 2023) | | | ✓ | | | ✓ |
| FVMVC (Sheng et al., 2023) | ✓ | | ✓ | | | ✓ |
| HearFace (Lee et al., 2024a) | | | ✓ | | | ✓ |
| VoxEditor (Sheng et al., 2024) | ✓ | | | ✓ | | ✓ |
| **UniSpeaker (Ours)** | ✓ | ✓ | ✓ | ✓ | ✓ | ✓ |

# 1 INTRODUCTION

In recent years, the field of speech synthesis has seen remarkable progress, driven by innovations in generative models and the expansion of training data. Some models (Wang et al., 2023a; Du et al., 2024; Ju et al., 2024; Vyas et al., 2023) can clone a voice using only a few seconds of reference speech, achieving a level of naturalness and speaker similarity that closely resembles actual recordings. Despite significant advances in voice cloning, zero-shot speech synthesis still faces limitations in certain scenarios, such as providing voiceovers for artificially created virtual characters, where obtaining ideal reference speech is very difficult or even non-existent (Guo et al., 2023). Compared to reference speech, natural text descriptions offer a more user-friendly approach to express intentions for voice generation (Leng et al., 2024), and facial images, which are easier to obtain, also have a strong correlation with voice characteristics (Goto et al., 2020; Oh et al., 2019). In the absence of reference speech, utilizing other modalities allows for more flexible and convenient control over voice characteristics. Hence, multimodal voice description-based speech generation, which involves generating corresponding voice characteristics from natural text descriptions, face images, or other modalities, presents a promising approach.

Recently, several studies have explored text prompt-based voice control for speech generation. These research (Shimizu et al., 2024; Zhang et al., 2023) have developed internal voice description sets and use BERT (Devlin et al., 2019) networks to extract text embeddings for voice control. Some research (Lu et al., 2021; Sheng et al., 2023) aligns facial recognition representations with speaker embeddings and uses these facial representations for voice control. In addition to the aforementioned absolute voice descriptions, VoxEditor (Sheng et al., 2024) introduces the relative descriptions for voice attributes editing, allowing for more nuanced control over voice characteristics.

Despite notable advancements made in these studies, they still have limitations in two key aspects: (1) **The versatility of voice control**: Current methods often explore different voice description modalities and generation approaches independently. On the one hand, it faces challenges in handling diverse inputs (as shown in Table 1). On the other hand, it fails to combine multimodal voice descriptions for more fine-grained and unique voice generation. (2) **The performance of voice control**: Previous methods were typically trained from scratch on limited paired multimodal data (Zhang et al., 2023; Shimizu et al., 2024), resulting in sparse coverage of the voice space. Effective strategies for fusing multimodality to enrich this space remain unclear. Additionally, these methods often generate speech with voice characteristics that do not align well with the input voice descriptions. While large-scale speech generation models (Wang et al., 2023a; Shen et al., 2024; Du et al., 2024) demonstrate exceptional voice control capabilities, the scalable integration of multimodality to enrich the capabilities of these pre-trained models remains unexplored.

To address these limitations, we present **UniSpeaker**, a speech generation model that aligns multiple modalities into a consistent voice space through a unified framework. As illustrated in Figure 1, speaker embeddings and semantic tokens serve as key representations to generate speech. Speaker embeddings control the voice characteristics and can be extracted from various inputs. Semantic tokens convey the content and prosody of the generated speech, derived from either source speech or content prompt. To effectively integrate multimodal input for voice control, pre-trained modality-specific encoders extract corresponding representations from different modalities, and then a unified multimodal voice aggregator (MVA) aligns these multimodal representations into a consistent voice space. The MVA is built upon the designed KV-Former, a streamlined variant of the Transformer (Vaswani et al., 2017). The KV-Former leverages a set of learnable key-value vectors to build a shared multimodal voice space, where multimodal representations serve as queries. To improve the alignment between voice characteristics and other modalities, soft contrastive learning (SoftCL) is applied to relax the strict one-to-one contrastive constraint and leverage intra-modal discriminative information for guidance. Considering the advantages of supervised semantic-tokens, we use the open-source CosyVoice (Du et al., 2024) as the backbone for UniSpeaker. Prior to integrating multimodal voice descriptions, we employ the simple yet effective self-distillation (Anastassiou et al., 2024) to improve the voice disentanglement of the pre-trained CosyVoice, thereby preserving its general capabilities across different tasks.

Due to the lack of publicly available benchmarks for evaluating multimodality-driven voice control, we established a multimodality-based voice control (MVC) benchmark, encompassing five tasks: face-driven voice conversion (FaceVC), face-driven personalized text-to-speech (FaceTTS), text description-driven voice conversion (TextVC), text description-driven personalized text-to-speech (TextTTS), attribute-driven voice editing (AVE). Following previous works (Sheng et al., 2023; Yao et al., 2024a), MVC benchmark evaluates generated speech using multimodal voice descriptions across three aspects: voice suitability, voice diversity and speech quality. We evaluate UniSpeaker using the MVC benchmark, where it demonstrates superior performance on the aforementioned five tasks compared to previous modality-specific models. Speech samples are available at `https://UniSpeaker.github.io`.

## 2 RELATE WORK

### 2.1 LARGE SPEECH GENERATION MODELS

As speech generation systems (Tan et al., 2022; Kim et al., 2021) have achieved remarkable levels of naturalness and robustness, recent research (Ju et al., 2024; Lee et al., 2024b) has shifted focus towards exploring novel generative models, advanced modeling objectives, and larger-scale datasets to pursue voice diversity. When integrating multimodal voice descriptions, it is crucial to preserve the performance of pre-trained speech generation models in terms of naturalness, robustness, and prosody. Some representative large-scale speech generation (Wang et al., 2023a; Kim et al., 2024; Chen et al., 2024) models typically leverage a neural codec to convert speech waveforms into discrete acoustic token sequences, along with an autoregressive language model to generate discrete tokens from text. However, the discrete acoustic token sequences entangle content, speaker, and prosodic information in this approach, complicating the alignment of multimodal voice characteristics without disrupting the content and prosody of the generated speech. Recently, CosyVoice (Du et al., 2024) has utilized supervised semantic tokens (Radford et al., 2023) as the modeling objectives for a large language model (LLM). Subsequently, a conditional flow matching model (CFM) generates speech based on semantic tokens, speaker embeddings and mel spectrograms prompt. Since the semantic tokens primarily encompass content and prosodic information, the speaker information included is limited. This facilitates further voice disentanglement and the integration of multimodal voice descriptions, making CosyVoice well-suited as the backbone for the UniSpeaker model proposed in this paper.

### 2.2 MULTIMODALITY-DRIVEN VOICE CONTROL FOR SPEECH GENERATION

Modeling diverse voice characteristics has consistently been a critical focus in the field of speech synthesis. Recent works, such as PromptTTS2 (Leng et al., 2024), Audiobox (Vyas et al., 2023), and others (Guan et al., 2024; Yang et al., 2024; Ji et al., 2024), have explored using text prompts to control the style or emotion of generated speech. However, only a few studies have specifically

targeted voice control with text prompt (Shimizu et al., 2024; Zhang et al., 2023). Text prompt-based style control TTS methods typically convert speech attributes like pitch, energy, duration, and emotion into natural style prompts using LLMs. Since these style prompts primarily reflect prosody and capture minimal speaker individuality, achieving the desired voice control remains challenging.

In the field of multimodal voice control, researchers have previously attempted to align different voice description modalities with speaker embeddings using models such as memory networks (Sheng et al., 2023), mixture density networks (Shimizu et al., 2024), and latent diffusion (Yao et al., 2024a), as well as loss functions like MSE loss (Lu et al., 2021), cosine similarity loss (Zhang et al., 2023), and perceptual loss (Weng et al., 2023). However, these alignment methods relied on parallel datasets and were challenging to extend directly to additional modalities. Performance-wise, previous face-based methods (Lee et al., 2023) generally ensured gender accuracy but often produced incongruous voice characteristics, such as generating a youthful voice for an elderly face. Additionally, VoxEditor (Sheng et al., 2024) is limited to performing voice attribute editing on existing source speech, thus offering restricted voice diversity. In response, the proposed UniSpeaker employs a unified voice aggregator to construct a shared voice space that can be easily extended to new modalities, achieving versatile and diverse voice control.

## 3 METHODS

In this section, we first review the backbone CosyVoice, then introduce how multimodal voice descriptions are integrated into a pre-trained speech generation model, and finally outline our training strategy with SoftCL and self-distillation.

### 3.1 PRELIMINARIES

Our backbone leverages supervised semantic tokens (Radford et al., 2023; Ye et al., 2024) as modeling objectives, utilizing an LLM for text-to-token generation and a CFM for token-to-speech synthesis. Given a dataset $\mathcal{D} = \{\mathbf{x}_i, \mathbf{y}_i\}$, where $\mathbf{x}$ is a speech sample and $\mathbf{y}$ is the corresponding text transcription, the sequence input to the LLM is mainly comprised of $\{\mathbf{s}, \mathbf{Y}, \mathbf{C}\}$, where $\mathbf{s}$ represents the speaker embeddings of $\mathbf{x}$, $\mathbf{Y}$ is the text embedding of $\mathbf{y}$ and $\mathbf{C}$ is the semantic tokens of $\mathbf{x}$. The LLM is then trained in an autoregressive manner to minimize the negative log-likelihood of semantic tokens $\mathbf{C}$. The core of CFM is to construct a probability density path from a prior distribution to $p_0(\mathbf{X})$ to the data distribution of the Mel-spectrograms $q(\mathbf{X})$. The probability density path is defined by a time-dependent vector field $\mathbf{v}_t(\mathbf{X})$, which generates the flow $\phi_t$ through an ordinary differential equation (ODE). The flow matching model is trained using optimal-transport conditional flow matching (OT-CFM) (Tong et al., 2023), which can be written as follows,

$$\mathcal{L}_{\text{OT-CFM}} = \mathbb{E}_{t, p_0(\mathbf{X}_0), q(\mathbf{X}_1)} \left| \omega_t(\phi_t^{OT}(\mathbf{X}_0, \mathbf{X}_1)|\mathbf{X}_1) - \nu_t(\phi_t^{OT}(\mathbf{X}_0, \mathbf{X}_1)|\theta_{CFM}) \right|, \tag{1}$$

where $\phi_t^{OT}(\mathbf{X}_0, \mathbf{X}_1) = (1-t)\mathbf{X}_0 + t\mathbf{X}_1$ and $\omega_t(\phi_t^{OT}(\mathbf{X}_0, \mathbf{X}_1)|\mathbf{X}_1) = \mathbf{X}_1 - \mathbf{X}_0$. The speaker embeddings $\mathbf{s}$, speech tokens $\mathbf{C}$, and masked Mel-spectrogram prompt $\tilde{\mathbf{X}}_1$ are also fed into the neural network to match the vector field with learnable parameters $\theta_{CFM}$,

$$\nu_t(\phi_t^{OT}(\mathbf{X}_0, \mathbf{X}_1)|\theta_{CFM}) = \text{NN}\left(\phi_t^{OT}(\mathbf{X}_0, \mathbf{X}_1), t; \mathbf{s}, \mathbf{C}, \tilde{\mathbf{X}}_1\right). \tag{2}$$

Therefore, both the LLM and CFM receive speaker embeddings. Our preliminary experiments revealed the CFM plays a primary role in voice control, while the LLM still exerts some influence (details in Appendix A). To improve voice disentanglement, self-distillation is applied to the pre-trained CFM, permitting multimodal voice descriptions to be integrated exclusively in the CFM.

### 3.2 MULTIMODAL VOICE DESCRIPTION INTEGRATION

We incorporate multiple modalities into the self-distillation CFM model, allowing various inputs to control the voice characteristics of generated speech. As shown in Figure 2, each modality is first processed by a pre-trained, modality-specific encoder to obtain the corresponding feature. Each kind of feature is then transformed into a latent vector via adaptive average pooling or a multi-layer perceptron (MLP) (Yao et al., 2024b). Those vectors across modalities are mapped into a unified voice space through a shared MVA, producing the corresponding speaker embeddings. These speaker embeddings are then fed into the CFM for speech generation.

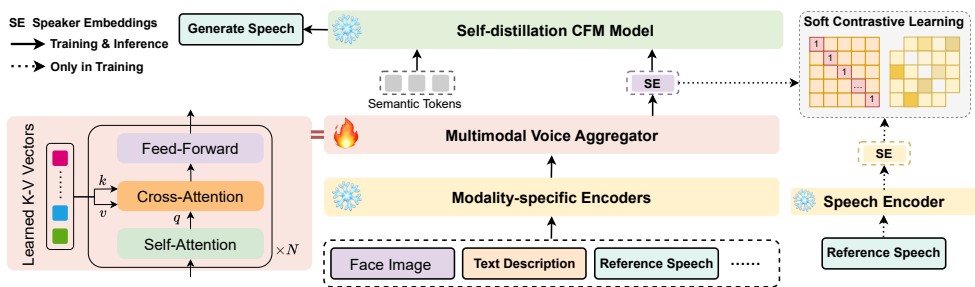

Figure 2: Overview of multimodal voice description integration.

**Modality-Specific Encoders** UniSpeaker employs three modality-specific encoders to process face images, speech and text. For face images, we use the MTCNN (Zhang et al., 2016) model for face detection, followed by a pre-trained convolutional neural network-based face recognition model (Schroff et al., 2015; Liu et al., 2022) to obtain global representations $\mathbf{s}_f$. For text description, T5 model (Raffel et al., 2020) is utilized to extract variable-length representations $\mathbf{s}_t$. For reference speech, the pre-trained speaker verification network (Wang et al., 2023b) from open-source CosyVoice is leveraged to extract speaker embeddings $\mathbf{s}_r$. More details about voice attribute descriptions are provided in Appendix B.

**Multimodal Voice Aggregator** Then global representations of different modalities should be aligned with speaker embeddings within the voice space. Previous methods relied on limited datasets that matched only two modalities for alignment, resulting in a sparse distribution in the voice space and weak generalization capabilities. Therefore, effectively utilizing multimodal voice descriptions through joint modeling to share a unified voice space, and thereby enhance the performance of each modality, remains an open question.

Inspired by Q-Former (Li et al., 2023b) and the memory mechanism (Sheng et al., 2023; Lee et al., 2021), we propose the KV-Former architecture as a unified multimodal voice aggregator. This architecture integrates learnable key-value vectors into a simplified Transformer, as shown in Figure 2. The multimodal representations act as queries and perform multi-head cross-attention with the learnable key-value vectors to retrieve the most informative representation in the voice subspace. The formulation of this process is as follows,

$$\mathbf{q} = \mathbf{W}^q \mathbf{s}_m, \mathbf{k} = \mathbf{W}^k \mathbf{f}, \mathbf{v} = \mathbf{W}^v \mathbf{f}, \mathbf{a}_m = \text{Softmax}\left(\frac{\mathbf{q}\mathbf{k}^T}{\sqrt{d}}\right)\mathbf{v}, \tag{3}$$

where $\mathbf{W}$ are the projection matrices in attention, $\mathbf{s}_m \in \{\mathbf{s}_f, \mathbf{s}_r, \mathbf{s}_t\}$ represents various state vectors, $\mathbf{f}$ are learnable key-value vectors, $d$ is the dimension of $\mathbf{f}$, and $\mathbf{a}_m$ is the output of cross attention. In this process, the learnable key-value vectors create an information bottleneck, interacting with the three modalities to build a shared voice space. Additionally, MVA adopts a speech-anchoring mechanism, reference speech is used as input for MVA with a 50% probability. In this way, even without parallel data between all modalities, different modalities achieves potential alignment in the voice space through shared k-v vectors and joint training. The inclusion of additional modality data can facilitate performance of current modality for voice control (evaluated in Section 5.4). Our module also allows for easy expansion to new modalities by adding the a modality-specific encoder.

To integrate multimodal inputs for voice control without losing the general abilities of CFM, we feed the output of MVA to the CFM and adapt the model without changing the CFM weights. The MVA is trained to optimize $\mathcal{L}_{\text{OT-CFM}}$ and Equation (2) is transformed as follows to fit speaker embeddings,

$$\nu_t(\phi_t^{OT}(\mathbf{X}_0, \mathbf{X}_1)|\theta_{MVA}) = \text{NN}\left(\phi_t^{OT}(\mathbf{X}_0, \mathbf{X}_1), t; \mathbf{v}_m, \mathbf{C}\right), \tag{4}$$

where $\mathbf{v}_m \in \{\mathbf{v}_f, \mathbf{v}_r, \mathbf{v}_t\}$ and $\mathbf{v}_f, \mathbf{v}_r, \mathbf{v}_t$ are the outputs of applying MVA to $\mathbf{s}_f, \mathbf{s}_r, \mathbf{s}_t$, respectively. In this manner, CFM can integrate multiple modalities for voice control and keep its ability to generate natural and robust speech.

### 3.3 TRAINING STRATEGIES

**Soft Contrastive Learning** Relying solely on OT-CFM to optimize MVA leads to slow convergence, and the generated speech may exhibit voice discordance with the input voice descriptions. Inspired by previous studies (Gao et al., 2024; Wang et al., 2024), we additionally introduce the SoftCL strategy for speech-anchoring multimodal alignment, including both inter-modal and intra-modal alignment, as shown in Figure 2. For inter-modal alignment, we employ InfoNCE (Radford et al., 2021), which pulls the paired multimodal and speaker embeddings closer together while pushing the unpaired ones apart. In addition, to bring cross-modal similarities closer to the distribution within each modality, intra-modal similarities serve as soft labels. Specifically, given a batch of $N$ multimodal-voice speaker embeddings pairs $\{(\mathbf{v}_m^i, \mathbf{s}_r^i)\}_{i=1}^N$, the intra-model self-similarity vector $p_i(\mathbf{s}_r, \mathbf{s}_r) = \{p_{ij}(\mathbf{s}_r, \mathbf{s}_r)\}_{j=1}^N$ can be obtained by:

$$p_{ij}(\mathbf{s}_r, \mathbf{s}_r) = \frac{\exp\left(\text{sim}\left(\mathbf{s}_r^i, \mathbf{s}_r^j\right)/\tau\right)}{\sum_{j=1}^N \exp\left(\text{sim}\left(\mathbf{s}_r^i, \mathbf{s}_r^j\right)/\tau\right)}, \tag{5}$$

where $\tau$ is a learnable temperature coefficient, initialized to 0.07, and $\text{sim}()$ denotes the dot product used to calculate similarity. Despite intra-model self-similarity, the confidence of positive samples still outweighs that of negatives, potentially overshadowing negatives in cross-modal relation alignment. To address this, we disentangle the negatives in the distribution to boost the relation alignment. For the self-similarity vector $p_i(\mathbf{s}_r, \mathbf{s}_r) \in \mathbb{R}^{1 \times N}$, the neg-disentangled $p_i^*(\mathbf{s}_r, \mathbf{s}_r) \in \mathbb{R}^{1 \times N-1}$ distribution is calculated as follows,

$$p_{ij}^* = \frac{\exp\left(p_{ij}\right)}{\sum_{k=1, k \neq i}^N \exp\left(p_{ik}\right)}. \tag{6}$$

We also apply the above negative disentanglement to $p_i(\mathbf{s}_r, \mathbf{v}_m)$, yielding $p_i^*(\mathbf{s}_r, \mathbf{v}_m)$. Then, the intra-modality alignment supervision can be achieved with negative disentanglement as follows,

$$\mathcal{L}_{\text{INTRA}} = \frac{1}{N} \sum_{i=1}^N \text{KL}\left(p_i^*(\mathbf{s}_r, \mathbf{s}_r) \| p_i^*(\mathbf{s}_r, \mathbf{v}_m)\right), \tag{7}$$

where KL represents the Kullback-Leibler Divergence. Generally, UniSpeaker is trained to optimize the following loss function,

$$\mathcal{L} = \mathcal{L}_{\text{OT-CFM}} + \lambda_1 \mathcal{L}_{\text{INTRA}} + \lambda_2 \mathcal{L}_{\text{INTER}}, \tag{8}$$

where $\mathcal{L}_{\text{INTRA}}$ is the InfoNCE loss, $\lambda_1$ and $\lambda_2$ are hyper-parameters used to balance each loss term.

**Self-distillation** Before integrating multimodal voice description, self-distillation is applied to fine-tune the CFM to improve voice disentanglement. Specifically, we utilize semantic tokens from the source speech, along with Mel-spectrogram prompt and speaker embeddings from another randomly selected speaker, and feed them into the CFM to perform voice conversion. This converted speech maintains the content and prosody of the source speech while almost entirely removing its speaker information (objective evaluations are shown in Table 5). Then, given the semantic tokens $\bar{\text{C}}$ of source speech and speaker embeddings $\mathbf{s}$ of converted speech as prompt, the CFM is fine-tuned to predict the source speech. Specifically, we removed the masked Mel-spectrogram prompt to improve the voice control by the speaker embeddings, transforming Equation (2) as follows,

$$\nu_t(\phi_t^{OT}(\mathbf{X}_0, \mathbf{X}_1)|\theta_{FM}) = \text{NN}\left(\phi_t^{OT}(\mathbf{X}_0, \mathbf{X}_1), t; \mathbf{s}, \bar{\text{C}}\right). \tag{9}$$

In this way, the voice characteristics of the generated speech is controlled by the speaker embeddings input to the CFM. This allows the integration of multimodal voice description directly into the CFM, simplifying the process without requiring modifications to the LLM.

## 4 DATASET AND BENCHMARK

Four modality-specific datasets were used to train the UniSpeaker. For the facial modality, we used the LRS3-TED dataset (Afouras et al., 2018), which includes TED Talks from 5,594 speakers, totaling approximately 400 hours of video. For the text description, LibriTTS-P (Kawamura et al.,

2024) was utilized with annotations for both voice characteristics and style, totaling approximately 585 hours of audio data from 2,443 speakers. Additionally, we collected speech-speaker identity description pairs from the internet, totaling about 90 hours. For the voice attribute modality, the VCTK-R (Sheng et al., 2024) dataset was selected, including pairwise comparisons of voice attributes among same-gender speakers, with 40 hours of audio data from 110 speakers.

The MVC Benchmark was proposed to evaluate multimodal voice control in five tasks, including FaceTTS, FaceVC, TextTTS, TextVC, and AVE. For face-related evaluation, we randomly selected 600 face images from the test set of LRS3-TED. In terms of textual descriptions, 600 sentences were randomly picked from the validation set and rewritten by a LLM (GPT-3.5-TURBO), ensuring that the meaning of the sentences remained unchanged. For voice attribute editing, following VoxEditor, 200 sentences were randomly selected from VCTK and edited on all attributes for evaluation. All above samples are unseen during training. More details can be found in the Appendix D.2.

The MVC benchmark evaluates the generated speech from three perspectives: voice suitability, voice diversity, and speech quality. Further details of the following metrics are in Appendix D.3.

1) **Voice suitability** evaluates whether the voice characteristics of the generated speech align with the input multimodal voice description. This includes three specific metrics: Speaker Similarity with Target (SST), Speaker Similarity Consistency (SSC), and MOS-Match. Speaker similarity can be computed by cosine similarity between the speaker embeddings, which are extracted from speech using a speaker verification model[1]. SST is measured by calculating the speaker similarity between the generated speech and reference speech of the target speaker. SSC assesses the consistency of the generated voice with various descriptions for the same speaker by calculating speaker similarity between the speech generated from different face images of the same speaker. MOS-Match is obtained through subjective listening tests for the mean opinion score to quantify how closely the voice characteristics of the generated speech align with the input description.

2) **Voice diversity** evaluates the model's ability to produce a diverse set of voice characteristics based on the descriptions of different speakers, rather than generating very similar ones. A metric named Speaker Similarity Diversity (SSD) is employed for evaluating voice diversity, which measures the speaker similarity between the speech generated from the descriptions of different speakers.

3) **Speech quality** assesses the robustness and naturalness of the generated speech, using two key metrics: word error rate (WER) and MOS-Nat. We employ an automatic speech recognition model[2] to transcribe the generated speech and compute the WER. MOS-Nat is determined through subjective listening tests for mean opinion scores to evaluate the naturalness of the generated speech.

## 5 EXPERIMENTS

### 5.1 EXPERIMENT SETTINGS

We trained the UniSpeaker using 4 NVIDIA TESLA V100 32G GPUs for 30K steps. The models were optimized using the AdamW optimizer with a learning rate of 1e-5 and a 10K warmup steps. The weights $\lambda_1$ and $\lambda_2$ in Equation (8) were set to 0.05. The model parameters for MVA are detailed in Table 7 of the Appendix. The speech tokenizer and codec LM were the same as those used in CosyVoice. For TTS, the codec LM accepted only text inputs without speaker embeddings.

We compared our UniSpeaker with 11 task-specific expert models in five tasks. We used the official code or pre-trained checkpoints of Imaginary Voice (Lee et al., 2023), FaceVC (Lu et al., 2021), SP-FaceVC (Weng et al., 2023), FVMVC (Sheng et al., 2023), and CosyVoice-Instruct (Du et al., 2024). For the other methods, we reproduced them according to their respective papers and evaluated them on the same dataset. Please refer to Appendix E for more details.

### 5.2 EVALUATION RESULTS

In this section, we conduct experiments comparing the UniSpeaker with the baselines and all objective and subjective evaluation results are reported in Table 2.

---

[1] https://github.com/modelscope/3D-Speaker
[2] https://huggingface.co/openai/whisper-large-v3

Table 2: Objective and subjective evaluation results of comparison systems. The definitions of all metrics can be found in 4. "-" denotes the results are not available.

| Task | Methods | Voice Suitability | | | Voice Diversity | Speech Quality | |
|---|---|---|---|---|---|---|---|
| | | SST ↑ | SSC ↑ | MOS-Match ↑ | SSD ↓ | WER ↓ | MOS-Nat ↑ |
| FaceTTS | Imaginary Voice(Lee et al., 2023) | 10.08 | 38.46 | 2.39 ± 0.09 | 32.17 | 8.23 | 2.45 ± 0.08 |
| | Face-StyleSpeech(Kang et al., 2023) | 11.02 | 37.09 | 2.78 ± 0.12 | 30.78 | 7.09 | 3.29 ± 0.10 |
| | SYNTHE-SEES(Park et al., 2024) | 10.97 | 38.81 | 2.92 ± 0.11 | 31.09 | 9.14 | 3.39 ± 0.09 |
| | UniSpeaker(Ours) | **12.48** | **40.75** | **3.18 ± 0.10** | **14.09** | **4.01** | **3.82 ± 0.08** |
| FaceVC | FaceVC(Lu et al., 2021) | 8.97 | 50.91 | 2.21 ± 0.11 | 30.19 | 10.90 | 2.79 ± 0.10 |
| | SP-FaceVC(Weng et al., 2023) | 9.52 | 52.29 | 2.39 ± 0.09 | 29.86 | 14.92 | 3.04 ± 0.10 |
| | FVMVC(Sheng et al., 2023) | 9.49 | 51.33 | 2.69 ± 0.07 | 22.60 | 11.94 | 3.31 ± 0.08 |
| | UniSpeaker(Ours) | **11.68** | **55.13** | **3.09 ± 0.10** | **15.91** | **4.98** | **3.80 ± 0.09** |
| TextTTS | PromptSpeaker(Zhang et al., 2023) | 17.39 | - | 3.64 ± 0.13 | 29.84 | 14.70 | 3.37 ± 0.10 |
| | Promptttts++(Shimizu et al., 2024) | 16.87 | - | 3.63 ± 0.12 | 35.42 | 15.08 | 3.41 ± 0.11 |
| | CosyVoice-Instruct (Du et al., 2024) | 14.51 | - | 3.71 ± 0.13 | 34.62 | 7.03 | **3.91 ± 0.09** |
| | UniSpeaker (Ours) | **23.09** | - | **3.85 ± 0.11** | **21.10** | **6.46** | 3.87 ± 0.13 |
| TextVC | PromptVC(Yao et al., 2024a) | 16.59 | - | 3.47 ± 0.07 | 36.98 | 7.08 | 3.64 ± 0.10 |
| | UniSpeaker(Ours) | **24.45** | - | **3.81 ± 0.09** | **24.04** | **6.29** | **3.77 ± 0.11** |
| AVE | VoxEditor(Sheng et al., 2024) | 41.48 | - | **3.78 ± 0.09** | 49.92 | 8.01 | 3.57 ± 0.10 |
| | UniSpeaker(Ours) | **49.04** | - | **3.79 ± 0.10** | **34.92** | **4.09** | **3.92 ± 0.09** |

In terms of voice suitability, our findings revealed that: 1) Across five tasks, UniSpeaker outperformed previous approaches on all three metrics, except for MOS-Match in the AVE task. While VoxEditor incorporates a complex residual memory network, the performance of our unified and scalable MVA remains competitive in MOS-Match. 2) In terms of face-based voice control, previous methods were generally effective in accurately controlling the gender of the voice characteristics but often exhibited obvious voice inconsistencies in subjective aspects such as age. In contrast, UniSpeaker achieved substantial improvements in both voice-age matching and overall subjective perception. 3) Additionally, we conducted an ABX test, as shown in Figure 5 of the Appendix, the voice characteristics generated by UniSpeaker sometimes can match the face image even more closely than those of the actual speaker. We encourage readers to listen to the samples on the demo page. 4) In text control, CosyVoice-instruct concatenates voice characteristic descriptions with the content prompt in the LLM without utilizing a pre-trained text prompt, resulting in difficulties grasping semantic information effectively and producing ambiguous voice characteristics. In contrast, UniSpeaker achieves excellent semantic-to-voice consistency, where similar semantics generate similar voice characteristics.

In terms of voice diversity, it is clear that UniSpeaker significantly outperforms previous methods across 5 tasks. Furthermore, we visualized the speaker embeddings of the generated speech from both SYNTHE-SEES and UniSpeaker systems using t-SNE (Chan et al., 2019), as shown in Figure 4 (a). The figure reveals that the voice space generated by our method is significantly richer, whereas the voice space of the baseline is relatively sparse. This indicates the voice characteristics generated by the baseline for different faces may being very similar, greatly limiting voice diversity.

In terms of speech quality, by freezing the CFM during training, UniSpeaker preserve the general abilities of our backbone. Consequently, UniSpeaker surpasses previous methods in overall speech quality, only the MOS-Nat slightly lags behind CosyVoice-Instruct. This lag is due to the CFM occasionally learning noise patterns from the dataset. Conversely, CosyVoice-Instruct only integrate multimodal voice descriptions in the LLM, resulting in minimal impact on speech quality.

## 5.3 ABLATION STUDY

Three ablation studies were conducted in our experiments. 1) To verify the effectiveness of MVA, the output of modality-specific encoders was mapped to the global representation, and it was directly fed into the CFM. 2) To assess the effectiveness of SoftCL, we removed the intra-class and inter-class contrastive losses from the output of MVA. 3) To validate the effectiveness of self-distillation, the performance of UniSpeaker and the open-source CosyVoice model (without self-distillation) was compared on TTS and VC tasks. We report the evaluation results for certain tasks in Table 3, with more evaluation results available in the Appendix F.

Table 3: The ablation study of UniSpeaker, measured by SST, SSD and SSC.

| Task | Methods | SST ↑ | SSD ↓ | SSC ↑ |
|---|---|---|---|---|
| FaceTTS | UniSpeaker | 12.48 | 14.09 | 40.75 |
| | w.o. MVA | 11.40 | 15.07 | 40.61 |
| | w.o. SoftCL | 11.57 | 15.94 | 38.28 |
| FaceVC | UniSpeaker | 11.68 | 15.91 | 55.13 |
| | w.o. MVA | 10.70 | 19.07 | 54.61 |
| | w.o. SoftCL | 11.08 | 19.24 | 51.55 |
| TTS | UniSpeaker | 44.30 | 10.03 | 33.32 |
| | w.o. self-distillation | 38.49 | 9.80 | 29.68 |
| VC | UniSpeaker | 39.37 | 10.34 | 50.64 |
| | w.o. self-distillation | 31.07 | 10.16 | 43.62 |

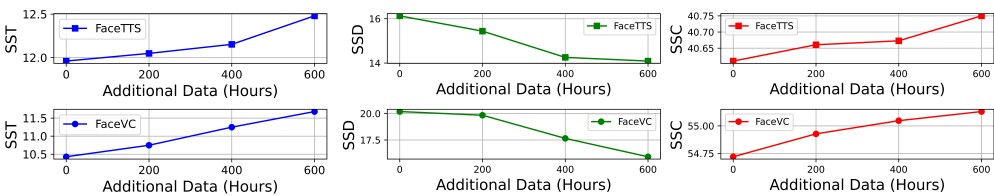

Figure 3: The evaluation results about different multimodal data scales on joint voice modeling. Here, the horizontal axis represents the amount of additional multimodal data, with "0" indicating that only the LRS3 dataset was used.

We have the following observations: 1) MVA proved beneficial for voice control with a shared multimodal voice space. It utilizes multimodal data for joint modeling through shared k-v vectors, resulting in a uniform distribution of the voice space. This promotes alignment between different modalities and enhances the model's performance in both voice diversity and voice suitability. 2) Removing SoftCL resulted in a decline across various metrics, specifically creating a significant mismatch between the generated voice and the input voice descriptions. 3) Eliminating self-distillation also had notable effects. Experimental results indicated that self-distillation significantly enhanced voice control, particularly in terms of SST. However, due to the limited data used for self-distillation, there was a slight reduction in voice diversity.

## 5.4 DISCUSSIONS

We investigated the impact of different multimodal data scales on the shared voice space. For face-driven voice control, we trained UniSpeaker using various datasets: solely LRS3, and additional datasets of varying sizes. The results, presented in Figure 3, show that increasing the amount of multimodal data improves the performance of FaceVC and FaceTTS, highlighting the benefits of multimodal joint modeling. Furthermore, the influence of additional multimodal data on SSC is less pronounced for SST and SSD, as SSC primarily relies on intra-modal relationships.

We randomly selected 8 unseen speakers and sampled 100 different face images from each for FaceTTS. The t-SNE visualization of speaker embeddings extracted from generated speech is presented in Figure 4 (b). We observed that for each speaker, the voice remained consistent across various facial images with different angles and backgrounds. This indicates that UniSpeaker demonstrates strong robustness to noisy information in facial images. Similarly, we used the LLM to rewrite the identity descriptions 60 times, ensuring consistent semantics with varied phrasing. Figure 4 (c) shows the visualization of the speech generated by TextTTS using these identity descriptions. The results indicate that for identity descriptions with the same semantics, the generated voices are consistent. Additionally, due to the presence of a multimodal shared space, UniSpeaker can accept multiple modalities simultaneously, allowing for more flexible and nuanced voice control. For details, please refer to the Appendix G and demo page.

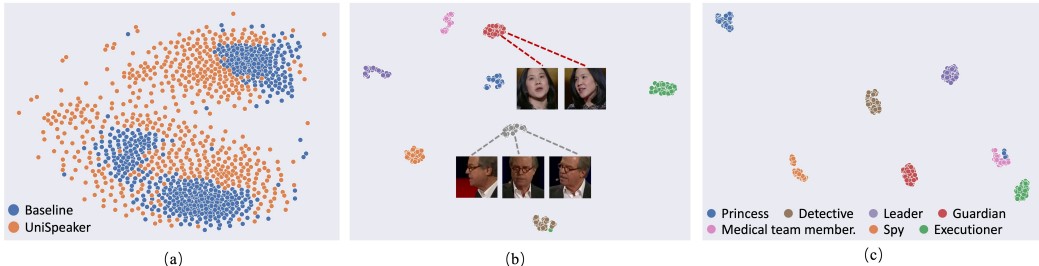

(a)                                (b)                                (c)

Figure 4: The visual analysis of UniSpeaker is presented here. Figure (a) uses t-SNE to visualize the voice space distributions of Baseline and UniSpeaker. In Figure (b), points of the same color represent the speech generated from different facial images of the same speaker. In Figure (c), points of the same color represent the speech generated from different identity descriptions of the same speaker, with the annotations serving as abbreviations of these text descriptions.

## 6 CONCLUSION

In this paper, we propose the UniSpeaker, a speech generation model that leverages multimodal voice description for voice control. Through a unified voice aggregator and designed training strategies, UniSpeaker outperforms previous modality-specific models across five tasks, generating voices that better match the input voice descriptions. In the future, we will explore how to more effectively utilize multiple voice descriptions of different modalities for one speaker simultaneously and apply our method on other more modalities for voice control.

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

Table 4: Ablation experiments to explore the impact of the LLM and CFM on voice characteristics under different conditions. The ✓ indicates the input is a regular speaker embeddings, while the ✗ denotes random noise input.

| Condition | LLM | CFM | SSIM |
|---|---|---|---|
| With Mel-spectrogram Prompt | ✓ | ✓ | 62.76 |
| | ✗ | ✓ | 57.03 |
| | ✓ | ✗ | 29.39 |
| | ✗ | ✗ | 24.04 |
| Without Mel-spectrogram Prompt | ✓ | ✓ | 44.07 |
| | ✗ | ✓ | 34.51 |
| | ✓ | ✗ | 8.44 |
| | ✗ | ✗ | 4.42 |

Table 5: Performance of different models on the voice conversion task, where * indicates the absence of Mel-spectrogram prompt. Note that these results are not comparable to those in Table 3 due to different test samples

| model | SSIM |
|---|---|
| Groud Truth | 69.67 |
| CosyVoice (Du et al., 2024) | 72.63 |
| CosyVoice* | 43.59 |
| FreeVC (Li et al., 2023a) | 36.31 |
| FACodec (Ju et al., 2024) | 52.73 |

## A ANALYSIS ABOUT COSYVOICE

### A.1 IMPACT OF THE LLM AND CFM MODULES ON VOICE CHARACTERISTICS

In the zero-shot speech synthesis task, the speaker embeddings input to either the LLM or Flow were replaced with random tensors of the same size. For evaluation, 500 sentences from the LRS3 dataset were selected, and the speaker similarity between the generated speech and the source speech was computed, as shown in Table 4. The results indicate that, compared to CFM, LLM has a significantly smaller impact on voice characteristics due to the limited voice characteristics contained in semantic tokens. Additionally, the balance between semantic and voice characteristics within semantic tokens across different scenarios is worth further exploration.

Additionally, by comparing the performance under both conditions in Table 4, we found that the Mel-spectrogram prompt carries more voice information than the speaker embeddings. In fact, the Mel-spectrograms offers a more detailed representation of voice characteristics, while the speaker embeddings provides a coarser one. For multimodal voice alignment tasks, multimodal voice descriptions are inherently incomplete and imprecise (Leng et al., 2024; Sheng et al., 2024), with a one-to-many mapping to voice characteristics. Thus, a coarse speaker embeddings is sufficient to serve as an anchor for multimodal alignment.

### A.2 PERFORMANCE OF COSYVOICE ON ZERO-SHOT VOICE CONVERSION

Before self-distillation, we evaluated the zero-shot voice conversion performance of CosyVoice. We extracted semantic tokens from the source speech and speaker embeddings, along with the Mel-spectrogram prompt from the reference speech, as inputs for the CFM. The generated speech retained the content and prosody of the source while altering the speaker's identity. We randomly selected 500 sentences from the LibriTTS test set to evaluate the performance of the CosyVoice, FreeVC[3] (Li et al., 2023a), and FAcodec[4] (Ju et al., 2024) models, with the experimental results presented in Table 5. During inference, when both the Mel-spectrogram prompt and speaker embeddings were provided, CFM-generated speech surpasses real speech in objective metrics. This

---

[3]https://github.com/OlaWod/FreeVC
[4]https://github.com/Plachtaa/FAcodec

is attributed to the fact that voice characteristics can exhibit local variations driven by content, rhythm, and emotion. This suggests that the audio produced by CFM is independent of the speaker information in the source semantic tokens, achieving exceptional disentanglement of voice characteristics. This makes it well-suited for self-distillation.

Without the Mel-spectrogram prompt, performance was inferior to FAcodec, which can be attributed to the inconsistency between training and inference, as the model was trained with both the Mel-spectrogram prompt and speaker embeddings as input. After self-distillation, the performance relying solely on speaker embeddings showed a marked improvement, as indicated in Table 3.

### A.3 PRELIMINARY EXPERIMENT ON FACE-BASED VOICE DESCRIPTION INTEGRATION

In our preliminary experiments, we directly integrated face embeddings into the CFM of the official CosyVoice[5]. Specifically, we utilized a pre-trained face encoder to extract the global face embeddings, replacing the speaker embeddings in the CFM as input. We evaluated the trained model on the zero-shot voice conversion task and found that the resulting SSIM score was around 4.8, indicating that the generated speech retained the identity information of the source speaker and did not achieve speaker conversion. This suggests that due to the cross-modal gap, CFM tends to extract voice information from the semantic tokens while neglecting the speaker information contained in the face. Therefore, to enable CFM to effectively utilize multimodal voice description integration, further voice disentanglement are necessary.

## B DETAILS OF VOICE ATTRIBUTE DESCRIPTIONS

For voice attribute description input, the model receives an input tuple consisting of two speech segments ($A$ and $B$) and a text description $t$. The text description states that A exhibits a certain attribute more prominently than B. For example, $t$ refers to "sounds more magnetic" meaning that voice characteritics of sample A is more magnetic than that of B. Following VoxEditor (Sheng et al., 2024), we first concatenate the speaker embeddings $\mathbf{s}_r^A, \mathbf{s}_r^B$ of two given speech samples and the text representation $\mathbf{s}_t$. Through MLP and Gaussian sampling, we predict the density difference $\alpha \in [0, 1]$ in attribute $x$ between the two speeches samples, and then obtain the target speaker embeddings $\mathbf{s}_r$ via linear interpolation: $\mathbf{s}_r = (1 - \alpha) \cdot \mathbf{s}_r^B + \alpha \cdot \mathbf{s}_t$. During inference, we can control the density of the target voice attribute by adjusting $\alpha$ within a range of 0 to 1.

## C DETAILS ABOUT INFONCE

Specifically, given a batch of $N$ multimodal-voice speaker embeddings pairs $\{(\mathbf{v}_m^i, \mathbf{s}_r^i)\}_{i=1}^N$, the multimodal-voice similarity vector $p_i(\mathbf{v}_m, \mathbf{s}_r) = \{p_{ij}(\mathbf{v}_m, \mathbf{s}_r)\}_{j=1}^N$ and voice-to-multimodal similarity vector $p_i(\mathbf{s}_r, \mathbf{v}_m) = \{p_{ij}(\mathbf{s}_r, \mathbf{v}_m)\}_{j=1}^N$ can be calculated as follows,

$$p_{ij}(\mathbf{v}_m, \mathbf{s}_r) = \frac{\exp\left(\text{sim}\left(\mathbf{v}_m^i, \mathbf{s}_r^j\right)/\tau\right)}{\sum_{j=1}^N \exp\left(\text{sim}\left(\mathbf{v}_m^i, \mathbf{s}_r^j\right)/\tau\right)}, p_{ij}(\mathbf{s}_r, \mathbf{v}_m) = \frac{\exp\left(\text{sim}\left(\mathbf{s}_r^i, \mathbf{v}_n^j\right)/\tau\right)}{\sum_{j=1}^N \exp\left(\text{sim}\left(\mathbf{s}_r^i, \mathbf{v}_n^j\right)/\tau\right)} \tag{10}$$

where $\tau$ is a learnable temperature coefficient, initialized to 0.07, and $\text{sim}()$ denotes the dot product used to calculate similarity. The inter-modal alignment loss is computed using cross-entropy as follows,

$$\mathcal{L}_{\text{INTER}} = \frac{1}{2N} \sum_{i=1}^N \mathcal{L}_{CE}\left(\mathbf{y}_i, p_i(\mathbf{v}_m, \mathbf{s}_r)\right) + \frac{1}{2N} \sum_{i=1}^N \mathcal{L}_{CE}\left(\mathbf{y}_i, p_i(\mathbf{s}_r, \mathbf{v}_m)\right) \tag{11}$$

where $\mathcal{L}_{CE}$ denotes the cross-entropy operation and $\mathbf{y}_i$ the one-hot label of $i_{th}$ pair.

---

[5]https://github.com/FunAudioLLM/CosyVoice

Table 6: An Example of using LLM to generate synonymous sentences.

| Diglogue | |
|---|---|
| LLM prompts: | Rewrite the following sentence, keeping the meaning unchanged, with a variety of sentence structures and styles. Please replace key words with synonyms: Princess X is honored as a priestess of the winter sea god, portrayed as a woman imbued with deep nostalgia and melancholy, while also being a contemporary fashion designer who cherishes traditional craftsmanship. |
| Response: | Princess X is revered as a high priestess of the deity of the winter sea, depicted as a figure filled with profound wistfulness and sorrow, yet she is also a modern fashion designer who values artisanal traditions. |

## D  DETAILS OF DATASETS AND BENCHMARK

### D.1  TRAINING DATASETS

For the LRS3-TED video dataset, 100 facial images per speaker were randomly selected from the videos, and a facial attribute detection model FairFace[6] was used to further clean the data. Specifically, the speaker's age and gender were estimated based on the 100 images, calculating the mean and variance. If the variance was too large, indicating poor video quality for that speaker, all samples from that speaker were discarded. Anomalies in these 100 images, often blurry pictures or images of a different speaker, were also filtered out. During training, a random image from the given speaker's image set was selected as input. FFmpeg[7] was used to extract 16kHz audio from the video. Additionally, the LRS3 dataset is also utilized for self-distillation of the CFM. For libritts-p, following promptts2 (Leng et al., 2024), we converted the these word-level annotations about voice characteristics into natural descriptive language using a language model.

### D.2  EVALUATION DATASETS

To evaluate the effectiveness of the text descriptions, we used a language model to rewrite sentences from the validation set while maintaining their original meaning, as shown in Table 6. This approach allows us to assess the model's generalization ability while providing targeted audio for comparison. Additionally, we prompted a large language model to randomly generate 100 character descriptions and voice characteristics descriptions, which can be considered out-of-domain. To further validate out-of-domain face image, we selected an Asian face dataset[8] for testing, given that the LRS3 dataset was collected from TED. The generated speech are available on the demo website.

### D.3  EVALUATION METRICS

For SST, when performing FaceTTS, FaceVC, TextTTS, and TextVC tasks, a multimodal voice description is provided along with a corresponding target speech. This allows us to directly calculate the speaker similarity between the generated speech and the target speech. However, for the AVE task, as there are no real voice characteristics, we calculate the speaker similarity with the source speech. The AVE task aims to edit specific voice attributes while preserving other characteristics as much as possible, so SST is used to assess whether the edited speech retains the original voice characteristics. Therefore, we need to combine SST and MOS-Match to comprehensively evaluate the performance of AVE.

For SSD, we matched generated speech with voice descriptions for different speakers to calculate speaker similarity, and then averaged the results across the evaluation dataset. A smaller average indicates greater voice diversity within the dataset. Specifically, for the AVE task, the diversity of the generated speech is assessed by applying the same voice attribute editing with the same weights to different speech inputs.

---

[6] https://github.com/dchen236/FairFace
[7] https://ffmpeg.org/
[8] https://github.com/X-zhangyang/Asian-Face-Image-Dataset-AFD-dataset

For SSC, pairwise matching of different images of the same person was performed to calculate their speaker similarity. These values were then averaged across the entire evaluation dataset. A higher average indicates greater voice similarity between different photos of the same individual, suggesting that the model is robust to background noise and other variations in the images.

For MOS-Match and MOS-Nat, subjective evaluation were conducted on Amazon Mechanical Turk[9]. Twenty sentences were randomly selected, and 20 listeners were asked to score each generated utterance on a scale from 1 (completely mismatched or completely unnatural) to 5 (completely matched or completely natural) for both metrics.

## E  COMPARATIVE METHODS

FaceTTS baselines:

- Imaginary Voice (Lee et al., 2023) is based on a score-based diffusion model, specifically Grad-TTS. Imaginary Voice used perceptual loss applied to the Mel-spectrograms to further align facial features and language.
- Face-StyleSpeech (Kang et al., 2023) proposes the disentangling of prosody and timbre, using facial features to control timbre and reference audio to control prosody. It also employs a contrastive learning to align facial and speaker embeddings.
- SYNTHE-SEES (Park et al., 2024) utilizes three types of losses—contrastive learning, speaker classification, and perceptual loss—to align face and speaker embeddings.

FaceVC baselines:

- FaceVC (Lu et al., 2021) employed a three stage training strategy, including face-voice reparameterization and facial-to-audio transformation, to align the face and voice characteristics.
- SP-FaceVC (Weng et al., 2023) first employed a bottleneck-free strategy for speech disentanglement. Then, multi-Scale discriminator and feature matching loss was proposed to improve performance.
- FVMVC Sheng et al. (2023) used FaceNet to extract general face embeddings and employ the memory net to align the face embeddings and speaker embeddings.

TextTTS baselines:

- PromptSpeaker (Zhang et al., 2023) annotated an internal dataset of speaker descriptions on LibriTTS-R. Building on this dataset, PromptSpeaker employed a pre-trained BERT network in conjunction with a Glow model to achieve alignment with speaker embeddings.
- Prompttts++ (Shimizu et al., 2024) integrated a BERT network with a Gaussian mixture model to predict speaker embeddings based on text descriptions, utilizing cosine loss for alignment.
- CosyVoice-Instruct (Du et al., 2024) concatenated the speaker's description before the text content in the LLm module of CosyVoice during training.

TextVC baseline:

- PromptVC (Yao et al., 2024a) utilized HuBERT and k-means clustering to represent semantic intermediate representations, and employed a diffusion model to predict style representations based on text input. Here, we replaced the dataset with ours to predict speaker embeddings using the diffusion model.

AVE baseline:

- VoxEditor (Sheng et al., 2024) first annotated a dataset describing timbre characteristics and utilized a residual memory network to accomplish the voice attribute editing.

---

[9]https://www.mturk.com/

Table 7: The detailed model configurations of MVA.

| Configuration | Value |
|---|---|
| Layer | 8 |
| Attention Dim | 768 |
| Attention Heads | 16 |
| Linear Dim | 2048 |
| Dropout | 0.1 |
| KV Size | 128 |

Table 8: The results of ablation studies on TextTTS and TextVC tasks, measured by SST, SSD.

| Task | Methods | SST ↑ | SSD ↓ |
|---|---|---|---|
| TextTTS | UniSpeaker | 23.09 | 21.10 |
| | w.o. MVA | 21.07 | 21.18 |
| | w.o. SoftCL | 22.57 | 34.51 |
| TextVC | UniSpeaker | 24.45 | 24.04 |
| | w.o. MVA | 21.50 | 24.26 |
| | w.o. SoftCL | 22.06 | 35.07 |

## F    FURTHER ABLATION STUDIES

We present the ablation experiment results for the TextTTS and TextVC tasks in Figure 8. This indicates that MVA and SoftCL are also beneficial for text-based timbre control. Additionally, we conducted ablation experiments on the size of learnable key-value vectors and the number of MVA layers, and found that within a certain range, the performance of voice control is not significantly affected, yet no clear patterns could be derived.

## G    FURTHER DISCUSSION

A unified voice space is constructed through a unified voice compressor. To validate the benefits of this shared space, voice interpolation on the speaker embeddings from different modalities is performed, allowing for manually adjusting the interpolation weights $\alpha$. As shown in Figure 6, we achieve voice control by interpolating the speaker embeddings obtained from face and textual descriptions. We observe that the voice characteristics vary as $\alpha$ changes, speech samples are available in the demo page.

By mapping multiple modalities to a unified voice space, we can leverage these different modalities to more comprehensively describe voice characteristics. Both face images and textual descriptions maintain a one-to-many relationship with the voice characteristics themselves. This means that given a face image or a textual description, the model can generate multiple matching voice characteristics. When both the target speaker's face and the textual voice description are input simultaneously, the generated voice characteristics that align with both modalities will better meet user expectations. Furthermore, we can editing specific voice attributes for more refined optimization. In the future, we will explore how to more finely utilize multiple modalities for voice control concurrently.

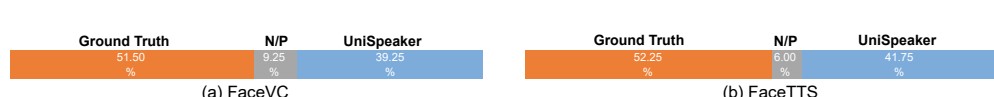

(a) FaceVC                                                  (b) FaceTTS

Figure 5: Average preference scores (%) of ABX tests about voice suitability in comparison, where participants were asked to select which of two speech samples—one generated based on the reference speaker's face image and one from the reference speaker's recording—better matched the speaker's appearance. "N/P" stands for "no preference". "Ground Truth" represents the real recording of the reference speaker.

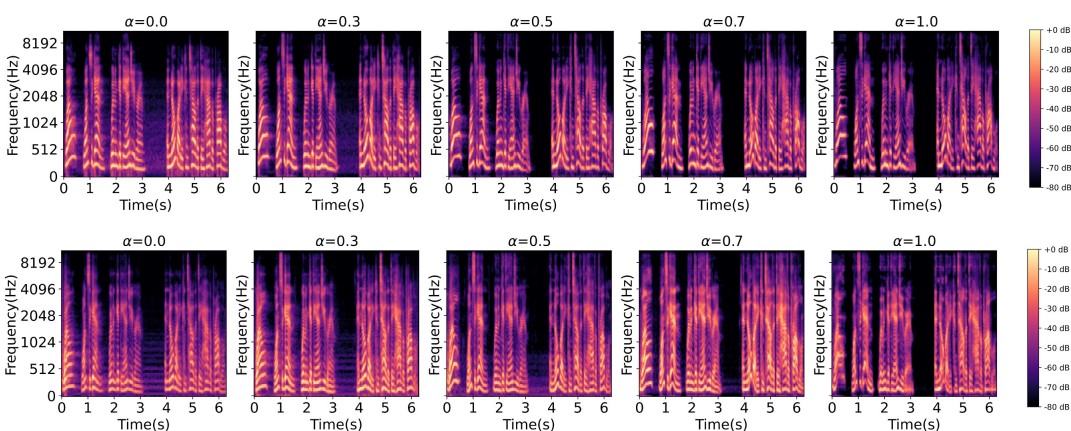

Figure 6: The voice characteristics controlled by both face and textual descriptions varies as $\alpha$ changes. When $\alpha = 0$, the voice characteristics are fully controlled by the face; when $\alpha = 1$, the voice characteristics are fully controlled by the textual description. We can observe the changes in voice characteristics and manually adjust $\alpha$ to achieve the desired voice characteristics.

