# OpenReview forum: "Unispeaker: A unified speech generation model for multimodality-driven voice control"
_ICLR.cc/2025/Conference — ICLR 2025 Conference Withdrawn Submission_

### Official Review · Reviewer_XE8n · 2024-10-27

**Soundness:** 2
**Presentation:** 3
**Contribution:** 2
**Rating:** 5
**Confidence:** 4

**Summary:**

The paper introduces UniSpeaker, a multimodal-driven speech generation model that integrates face images, text descriptions, voice attributes, and reference speech to enable detailed and flexible voice control. The model utilizes a KV-Former-based multimodal voice aggregator (MVA) to map these diverse input forms into a shared voice latent space, thus ensuring alignment between the generated voice and input descriptions. To support effective multimodal control, a soft contrastive learning technique is applied to support flexible multimodal alignment, further enhancing the model's capacity for voice creation. UniSpeaker also employs self-distillation for improving voice disentanglement and preserving the pre-trained CosyVoice model's capabilities across tasks.

The paper also introduces the MVC benchmark to evaluate multimodal voice control across five tasks—evaluating voice suitability, diversity, and quality. Experimentally, UniSpeaker outperforms previous modality-specific models, establishing itself as a more versatile solution for multimodal voice synthesis.

**Strengths:**

1. The multimodality-based voice control (MVC) benchmark proposed in this paper is beneficial for future research.
2. Although the proposed multimodal voice aggregator (MVA) module are not highly innovative, it is effective in terms of the final experimental results.
3. The proposed soft contrastive learning (SoftCL) and self-distillation strategies are intuitive and interesting.

**Weaknesses:**

The main weaknesses of this paper are a lack of clarity in some experimental setups, specifically as follows:

1. In Table 2, comparisons between the UniSpeaker and baselines are not fair enough. The official checkpoints of these baselines may be trained on different datasets. For example, on the TextTTS task, the training and testing sets in this paper adopt academic reading-style datasets like VCTK. However, CosyVoice-Instruct [1] uses an internal commercial dataset, which creates a domain gap with the MVC benchmark used in this paper. Therefore, the experimental results are not convincing. Perhaps a fairer comparison could be made by using the CosyVoice-Instruct model fine-tuned on the same training dataset. I can also understand that many baselines use non-open-source data, making it difficult to fully reproduce them on the same datasets used in this paper. At least, the authors should include the specific training datasets used for these baseline models in Table 2.

There are also some minor weaknesses:

2. In terms of the proposed dataset and benchmark, this paper primarily collects several open-source datasets to form its dataset and establishes the MVC benchmark, so the contribution is relatively modest. But I think the MVC benchmark could serve as a valuable tool for evaluating multimodal-driven voice control. In terms of novelty, this paper seems to integrate existing algorithms to propose a unified model, lacking some insightful algorithms or results.

3. Instead of using the weights $\alpha$ for interpolating the speaker embeddings obtained from face and textual descriptions, the multi-condition classifier free guidance (See Section 3.2.1 in [2]) may be a better choice for multi-modal voice control.

4. Using only a global speaker embedding may limit timbre reconstruction quality, as time-averaged pooling removes some time-varying timbre details. To address this, prior works like NaturalSpeech 3 [3] have incorporated both global speaker embeddings and detailed embeddings for enhanced speech reconstruction. The performance could be further improved by incorporating finer-grained speaker embeddings.

5. Additional details on the inter-modal and intra-modal alignment learning process may be included in Figure 2 to enhance clarity.

6. Typo: in Appendix F, line 1049, ``Figure 8`` -> ``Table 8``; in the title of this submission, ``Unispeaker`` -> ``UniSpeaker``.

7. This paper advances the development of multimodal-driven speech synthesis, a technology that raises certain ethical issues for society. Therefore, an ethics statement should be added to the end of the article.

Overall, as mentioned in Weakness 1, the training sets of different models are different, making the experimental results not promising enough. Therefore, I rate it 5. If there are any errors or inappropriate aspects in the points I've raised, please feel free to contact me at any time.

[1] Du, Zhihao, et al. "Cosyvoice: A scalable multilingual zero-shot text-to-speech synthesizer based on supervised semantic tokens." arXiv preprint arXiv:2407.05407 (2024).
[2] Brooks, Tim, Aleksander Holynski, and Alexei A. Efros. "Instructpix2pix: Learning to follow image editing instructions." Proceedings of the IEEE/CVF Conference on Computer Vision and Pattern Recognition. 2023.
[3] Ju, Zeqian, et al. "Naturalspeech 3: Zero-shot speech synthesis with factorized codec and diffusion models." arXiv preprint arXiv:2403.03100 (2024).

**Questions:**

My questions are included in the weaknesses part.

---

### Official Review · Reviewer_hkuv · 2024-10-31

**Soundness:** 3
**Presentation:** 3
**Contribution:** 2
**Rating:** 3
**Confidence:** 4

**Summary:**

This paper introduces a zero-shot TTS system that leverages multimodal conditional control, encompassing face images, text descriptions, reference speech, and voice attribute descriptions. Building upon CosyVoice, the paper proposes a contrastive learning-based approach to align different modalities with speaker embeddings within the voice space. Additionally, it employs a self-distillation technique to mitigate the influence of speaker information in semantic codes on the flow-matching model.

**Strengths:**

1. This paper enhances the capabilities of zero-shot TTS by expanding multimodal control, incorporating elements such as face images and text descriptions.

2. Building on the principles of self-distillation from SeedTTS [1], this paper augments CosyVoice's flow matching for voice conversion, particularly through the use of speaker embeddings to control voice timbre.

3. The paper introduces a multimodality-based voice control (MVC) benchmark, which includes five tasks: FaceVC, FaceTTS, TextVC, TextTTS, and AVE. The proposed model, along with baseline models, is evaluated on this benchmark, demonstrating superior performance in terms of model capabilities.

[1] Anastassiou P, Chen J, Chen J, et al. Seed-TTS: A Family of High-Quality Versatile Speech Generation Models[J]. arXiv preprint arXiv:2406.02430, 2024.

**Weaknesses:**

First, I believe this work represents a commendable extension to the existing zero-shot TTS framework. However, the paper's most significant weakness lies in its lack of novelty and failure to introduce new insights to the community.

1. All methods employed are merely combinations of existing frameworks, many of which have already been utilized in the TTS domain. Moreover, the conclusions drawn do not offer any groundbreaking insights or provoke thought among peers in the field. For instance, the

2. The use of contrastive learning to align features from other modalities with speaker embeddings, as well as the two-stage approach (AR + diffusion/flow matching), have been employed in numerous TTS works (SeedTTS, CosyVoice, FireRedTTS, Tortoise-TTS, SoundStorm...).

3. At the results level, while the authors do surpass state-of-the-art solutions on their proposed benchmarks, there is no further breakthrough. For example, how well does the model adhere to out-of-domain text descriptions (e.g., generating speech as if spoken by a dragon from a life story)? Or, can the face images be from animations or non-realistic faces? (In fact, I suspect the results may not be satisfactory due to the lack of diversity in the training data).

**Questions:**

1. Has the author evaluated the model's performance on out-of-domain inputs (e.g., text descriptions significantly different from the existing distribution or face images of animated characters)?

2. Given that the framework of CosyVoice has been utilized, has the author attempted to fine-tune the model directly on CosyVoice? Since the data volume used in this work is considerably smaller than the training data of CosyVoice.

---

### Official Review · Reviewer_Xqf8 · 2024-11-01

**Soundness:** 3
**Presentation:** 2
**Contribution:** 1
**Rating:** 3
**Confidence:** 5

**Summary:**

The paper introduces a unified framework for cross-modal speaker embedding extraction and conditioning, addressing the challenge of generating consistent speaker representations across diverse input modalities - face images, text descriptions, voice attribute descriptions, and reference speech. The authors enhance the modal alignment through the implementation of a soft contrastive loss function, specifically designed to strengthen the correspondence between voice descriptions and generated speech characteristics. To ensure the model's dependence solely on extracted speaker embeddings, the authors employ a self-distillation technique utilizing a pretrained conditional flow matching (CFM) model. The work's empirical validation is supported by the authors' development of a multimodality-based voice control benchmark, which serves as a standardized evaluation framework for both the proposed method and baseline approaches.

**Strengths:**

* Originality: UniSpeaker is the first model that can condition on 4 modalities (ref speech, text, face, attribute) for both TTS and speech-to-speech conversion. It has also proposed the first benchmark for multimodality-based voice control.

* Quality: using the benchmark proposed in the paper, among all the baselines, UniSpeaker performs the best in terms of voice suitability, voice diversity and speech quality.

* Clarity: the paper is generally clear

* Significance: as the first model that unify 4 conditioning modalities, this work is of some significance.

**Weaknesses:**

* Originality: this paper lacks originality. The whole pipeline is just a naive combination of existing work: the backbone is CosyVoice, and the modality-specific encoders are all from previous existing pretrained models. The only module that needs to be trained is the multimodal voice aggregator, which is also based on memory mechanism. And the so-called self-distillation is just to adapt the CFM to condition only on the speaker embeddings, i.e. removing the mel-spectrogram prompt. In addition, the proposed benchmark is also just a repetition of existing evaluation methods, including WER, MOS, and speaker-embedding-based similarity metrics (CosyVoice has the exact same formulation).

* Missing experiments: since this is a model that can handle multiple conditioning modalities, experiments should be conducted to assess the performance of conditioning on different numbers of modalities. For example, how good is the performance of adding additional face information together with text descriptions compared to only text / only face?
Besides, the paper fails to point out the number of K-V vectors used in the experiments. This is a very important hyper-parameter, since it directly defines the codebook size. Moreover, experiments should also be conducted to assess the effect of different numbers of K-V vectors and the utilization rate. It would also be more interesting to show if the different learned K-V vectors actually represent different properties of the voice, like pitch, gender, age, etc.

* Writing accuracy:
  * line 290, for the shape of neg-disentangled distribution, instead of 1xN-1, it should be 1x(N-1)
  * line 304, instead of L_{INTRA}, it should be L_{INTER}, i.e. inter-modality loss is the InfoNCE loss
  * line 312, instead of “the CFM is fine-tuned to predict the source speech”, it should be “the CFM is fine-tuned to predict the converted
  * speech”, since the input speaker embedding is from the converted speech
  *.Table 2, SST, SSC, SSD, and WER should be all in the range of (0, 1). Add (%) in the table or caption

**Questions:**

As mentioned in the Weaknesses section, it would be helpful to add the experiments about multi-modality conditioning and the number of K-V vectors.

**Details Of Ethics Concerns:**

N/A.

---

### Official Review · Reviewer_QGts · 2024-11-06

**Soundness:** 3
**Presentation:** 4
**Contribution:** 4
**Rating:** 8
**Confidence:** 4

**Summary:**

The paper addresses multi-modal voice control in speech generation, leveraging inputs from facial images, text, voice attribute prompts, and speech signal prompts. The proposed model integrates various types of speaker information to learn a unified voice-type space, which is subsequently applied in a downstream flow-matching network. Additionally, the paper introduces a benchmark to support future research on this task. Experimental results demonstrate that multi-modal control can enhance the flexibility and precision of controllable speech generation.

**Strengths:**

- The proposed UniSpeaker model offers comprehensive support for multiple voice control modalities, representing a substantial extension beyond similar models. Importantly, it supports both voice conversion (VC) and text-to-speech (TTS) tasks.
- The benchmark introduced in the paper could serve as a valuable resource for the research community.
- The UniSpeaker framework incorporates several notable technical innovations, including the use of KV-Former as a unified multi-modal voice aggregator, soft contrastive loss, and self-distillation. Each of these techniques is validated through extensive ablation studies.

**Weaknesses:**

The paper presents a well-rounded method and extensive experimental analysis. While I have only minor concerns (detailed in the question section), no significant issues were identified in the current draft.

**Questions:**

- MOS Evaluation: The mean opinion score (MOS) evaluation demonstrates only a modest significance level, likely due to a limited number of samples and listeners. Expanding the sample size could increase the statistical significance of these evaluations.
- Bi-Modal Ablation Study: Although the work shows that combining all modalities enhances performance, further ablation studies on bi-modal combinations could help readers understand the complementary effects between specific modality pairs.
- SSD Evaluation Justification: The SSD evaluation lacks sufficient justification, as the speaker embeddings are not optimized for this purpose. Subjective analysis might provide a more appropriate fit for this evaluation, offering a more accurate reflection of perceptual outcomes.

**Details Of Ethics Concerns:**

The benchmark involved in the paper is highly related to speaker identity-related studies. The proper source and usage of the data could potentially need additional ethics review.

---

### Note · Authors · 2024-11-13

**Comment:**

Thank you very much for the comments and suggestions from the reviewers. We have decided to withdraw this paper.

**Withdrawal Confirmation:**

I have read and agree with the venue's withdrawal policy on behalf of myself and my co-authors.